# Diagnostic Challenges in Neonatal Respiratory Distress—Congenital Surfactant Metabolism Dysfunction Caused by *ABCA3* Mutation

**DOI:** 10.3390/diagnostics12051084

**Published:** 2022-04-26

**Authors:** Justyna Rogulska, Katarzyna Wróblewska-Seniuk, Robert Śmigiel, Jarosław Szydłowski, Tomasz Szczapa

**Affiliations:** 1II Department of Neonatology, Neonatal Biophysical Monitoring and Cardiopulmonary Therapies Research Unit, Chair of Neonatology, Poznan University of Medical Sciences, 60-535 Poznan, Poland; justyna.latacz162@gmail.com (J.R.); tszczapa@ump.edu.pl (T.S.); 2Department of Family and Paediatric Nursing, Wroclaw Medical University, 50-996 Wroclaw, Poland; robert.smigiel@umw.edu.pl; 3Department of Otolaryngology, Head and Neck Surgery and Laryngological Oncology, Poznan University of Medical Sciences, 61-701 Poznan, Poland; jszydlow@ump.edu.pl

**Keywords:** inherited surfactant deficiency, neonatal respiratory distress, *ABCA3* gene, neonatal respiratory failure, congenital surfactant defects

## Abstract

Surfactant is a complex of phospholipids and proteins produced in type II pneumocytes. Its deficiency frequently occurs in preterm infants and causes respiratory distress syndrome. In full-term newborns, its absence results from mutations in the *SFTPC, SFTPB, NKX2-1*, or *ABCA3* genes involved in the surfactant metabolism. *ABCA3* encodes ATP-binding cassette, which is responsible for transporting phospholipids in type II pneumocytes. We present a case of a male late preterm newborn with inherited surfactant deficiency in whom we identified the likely pathogenic c.604G>A variant in one allele and splice region/intron variant c.4036-3C>G of uncertain significance in the second allele of *ABCA3*. These variants were observed *in trans* configuration. We discuss the diagnostic challenges and the management options. Although invasive treatment was introduced, only temporary improvement was observed. We want to raise awareness about congenital surfactant deficiency as a rare cause of respiratory failure in term newborns.

## 1. Introduction

Surfactant plays a crucial role in reducing surface tension in alveoli to maintain stable air space for gas exchange and prevent end-expiratory alveolar collapse. Its deficiency is mainly observed in premature neonates, as it may lead to respiratory distress syndrome. Congenital surfactant deficiency caused by mutations in various genes playing an important role in surfactant biosynthesis may cause a similar presentation of severe respiratory distress syndrome with lethal respiratory failure in full-term newborns or interstitial lung disease (ILD) in older children and adults [1,2]. Congenital surfactant deficiency is associated with mutations in the *SFTPC*, *SFTPB*, *NKX2-1*, or *ABCA3* as well as *CSF2RA, CSF2RB, SFTPA1, SFTPA2, SFTA3,* and *SFTPD* genes involved in the surfactant biosynthesis [3,4].

Surfactant is a complex of highly specific phospholipids and proteins synthesized by epithelial cells called type II pneumocytes. It is intracellularly embedded in inclusion organelles-lamellar bodies. Phospholipids make up almost 90% of surfactant weight; the remaining 10% are hydrophobic proteins SP-B and SP-C and hydrophilic proteins from the collectin family SP-A and SP-D [5]. The specific surfactant proteins A (SP-A), B (SP-B), C (SP-C), and D (SP-D) are encoded by the *SFTPA, SFTPB, SFTPC*, and *SFTPD* genes and have considerable functional significance [6]. Type II pneumocytes differentiate between weeks 24 and 34 of gestation. Premature newborns with RDS have only about one-tenth the amount of surfactant compared to healthy full-term newborns [6]. Mutations of *SFTPB, SFTPC*, and *ABCA3* genes can lead to qualitative and quantitative surfactant defects [2]. However, the most common cause of primary surfactant defect is the loss of function mutations in the *ABCA3* gene [7].

*ABCA3* is a member of the ATP-binding cassette (ABC) transporter family, which encodes membrane proteins that transport compounds across biologic membranes. It is a 1704-amino-acid protein expressed mainly in the alveolar epithelial cells at the limiting membrane of lamellar bodies [7]. It has been detected from 26–27 weeks of gestation in normal fetuses [4]. The *ABCA3* is responsible for transporting phospholipids into the lamellar bodies (LBs) in type II pneumocytes. It is crucial for AT2 cells’ homeostasis, lipid composition, and protein maturation of surfactant. There have already been more than 150 patients with recessive mutations identified in the *ABCA3* gene that may lead to respiratory distress in full-term newborns and older children [8]. Bi-allelic *ABCA3* mutations were the most frequent cause of congenital surfactant deficiency [6].

We report a case of a late preterm infant with severe respiratory failure from the first day of life, in whom we identified *ABCA3* variants *in trans* c.604G>A and c.4036-3C>G. These variants were detected in the mother and father of the presented child in a heterozygous state. *ABCA3* variants in both alleles may cause a loss of function of the *ABCA3* protein and significant surfactant deficiency with severe respiratory failure.

## 2. Case Report

The patient was a late-preterm baby boy, delivered normally at the primary care center at 36 weeks of gestation, after an uncomplicated pregnancy, with a birth weight of 2097 g and Apgar scores of 10 in the 1st and the 5th minute of life. The parents do not have another child together. The mother has three healthy children with her previous partner.

Within the first few hours of his life, he developed severe respiratory distress and required respiratory support. At first, the nCPAP therapy was introduced, and the patient was transported to the tertiary care center. After several hours he required non-invasive ventilation, and since his status continued to worsen, mechanical ventilation was initiated on the 3rd day of life. Ultimately, due to increasing respiratory acidosis, he was put on high-frequency ventilation (HFO) on the 5th day of life. He also required inhaled nitric oxide from the 4th to the 15th day of life and later from the 23rd until the 30th day of life. The baby received two doses of exogenous surfactant during hospitalization on the 6th and 21st day of life without significant improvement. The corticosteroids (dexamethasone) were administered twice in 10-day courses (0.03 mg/kg twice a day) without any progress. Echocardiography excluded congenital heart defects. The mechanical ventilation was complicated by pneumothorax on the 23rd day of life, treated with suction drainage for seven days.

On pulmonary ultrasound, bilaterally compact B-artifacts were observed, which created the image of white lungs. Chest radiograms showed diffuse opacification of both lungs with air bronchograms. Computed tomography showed a characteristic pattern of interstitial lung disease with ground-glass opacities and pneumatocele. In the extended screening test, cystic fibrosis was excluded. The patient was transferred to the otolaryngology department to broaden diagnostics. The bronchofiberoscopy raised suspicion of a large tracheoesophageal fistula, which was eventually excluded after the consecutive examination (Appendix A).

The patient was admitted to the Department of Newborns’ Infectious Diseases on the 54th day of life. He was on HFO ventilation, FiO_2_ 100%. We performed the procedure of surfactant lung lavage as a rescue treatment, after which we observed transitional improvement, which enabled us to use conventional mechanical ventilation (SIMV) instead of HFO. However, FiO_2_ requirements and ventilator parameters remained very high, with oxygen demand of 1.0 and MAP of 16 cm H_2_O. On the 74th day of life, the patient presented with cardiac arrest after accidental extubation. He was resuscitated after 4 min of chest compressions and drugs administration.

The pulmonary ultrasound and chest radiographs performed frequently during the patient’s hospital stay did not change over time. Figure 1 presents pulmonary ultrasound on the 83rd day of life in which compact B-artifacts create the image of white lungs with many subpleural consolidations. The chest radiographs shown in Figure 2 are taken in exactly one month. On both X-rays, we can see diffuse reticular granularity and air bronchograms. Computed tomography repeated after several weeks (Figure 3 and Appendix A) showed again a pattern of interstitial lung disease with ground-glass opacities and pneumatocele.

In microbial culture from the bronchotracheal aspirate on the 67th day of life, Serratia marcescens was found. At the same time, profuse discharge from the airways was observed, and CRP was elevated and rising (from 11.72 up to 81.05 mg/L), so we diagnosed ventilator-associated pneumonia. The patient was treated with antibiotics for 21 days, but it did not cause any improvement in his general condition, although the inflammation markers returned to typical values.

Due to a very unstable state, the patient was given high doses of sedative drugs (Morphine 50 µg/kg/h and Midazolam 0.7 mg/kg/h). This strategy allowed us to change respiration parameters from SIMV with PIP approx. 40 cm H_2_O to PRVC with PIP approximately 30 cm H_2_O and PEEP of 12–14 cm H_2_O. We aimed to provide better comfort to the patient, so NAVA ventilation was introduced, which allowed lowering respiratory parameters, CO_2_ retention, and oxygen demand from 1.0 to 0.65–0.8 FiO_2_. The ventilator parameters on PRVC and NAVA ventilation are presented in Figure 4.

Genetic analysis of genes associated with surfactant metabolism disorders was performed during the hospital stay using next-generation sequencing. Informed consent was obtained from parents according to the local regulations. Genomic DNA was extracted from a peripheral blood EDTA sample. The most important genes involved in surfactant metabolism: *ABCA3, SFTPB*, and *SFTPC*, were analyzed with twist Bioscience Custom Panel in NextSeq Ilumina. No mutation was detected in the *SFTPC* and *SFTPB* genes; however, we identified the likely pathogenic c.604G>A variant in one allele and splice region/intron variant c.4036-3C>G of uncertain significance in the second allele of *ABCA3*. These variants were observed *in trans* configuration. Both parents are carriers of one variant each.

After the genetic result of the *ABCA3* variant related to surfactant dysfunction and in the absence of the possibility of causal treatment of the disease, the patient was transferred to the hospital closer to his family to introduce palliative care. Following discussion with parents, the “do not resuscitate” form was signed by specialists. The patient deceased at the age of 99 days.

## 3. Discussion

In term newborns, respiratory distress occurs in 5% to 7% of live births, mainly resulting from the abnormal transition from fetal to neonatal life. It is primarily manifested by tachypnea, nasal flaring, intercostal or subcostal retractions, audible grunting, and cyanosis. Most cases are mild and transient and are diagnosed as transient tachypnea of the newborn (TTN). Among other common causes, we distinguish persistent pulmonary hypertension of newborns (PPHN), meconium aspiration syndrome (MAS), and infectious conditions such as sepsis or pneumonia. However, severe respiratory distress is often due to non-pulmonary causes like congenital heart disease, air leaks, or pathoanatomic conditions of the pulmonary airway. Occasionally, term neonatal respiratory distress is associated with an inherited primary lung disease such as surfactant metabolism defects [9,10].

The *ABCA3* is a transmembrane protein found in pneumocytes type II lamellar bodies. It belongs to a more incredible family of ABC transporters, which mediate the transport of various physiologic lipid compounds. Mutations of their encoding genes are linked to several diseases, e.g., Stargardt’s disease (ABCA4), harlequin ichthyosis (ABCA12) [11]. *ABCA3* is a protein encoded by an 80 kb gene located on the 16th chromosome (16p13.3). The incidence of *ABCA3* mutation may vary between 1:4400 to 1:20,000 in the European and African populations [12].

*ABCA3* deficiency is one of the most frequent causes of genetic surfactant metabolism disorders [11] and should be suspected in neonates with severe neonatal respiratory distress syndrome refractory to conventional treatment [8]. It leads to abnormal lamellar bodies’ structure and abnormal surfactant lipid composition, associated with impaired processing of SP-B and SP-C proteins [13]. Surfactant isolated from infants with mutations in *ABCA3* presents decreased function and deficiency in phosphatidylcholine [14].

Qualitative functional characterization of *ABCA3* missense variants suggests 2 pathogenic classes: disrupted intracellular trafficking (type I mutant) or impaired ATPase-mediated phospholipid transport into the lamellar bodies (type II mutant). Each disease-associated *ABCA3* variant is associated with a diverse pulmonary phenotype [15].

The loss of *ABCA3* function leads to dramatic progress of neonatal distress syndrome and death within the first three months of life, whereas residual function is associated with milder disease [8]. The most common presentation of a baby with an *ABCA3* mutation that leads to respiratory failure is a full-term baby with moderate to severe respiratory distress and signs of diffuse lung disease without satisfactory history or laboratory findings [2,16]. The variants in *ABCA3* are unique to individuals and families and have a homozygous or compound heterozygous state. Heterozygous *ABCA3* missense variants are present in 1.5–3.7% of African and European descent individuals [12]. The most frequent mutation of the *ABCA3* gene in Europe is p. Glu292Val. It is carried by 1.3% of the Danish population [17]. This mutation is responsible for less than 10% of identified pathogenic alleles [18]. Most mutations are unique, with over 300 different variants reported in the literature, making genetic counseling difficult in the families, especially in the case of missense mutations (amino acid substitution) where phenotype-genotype correlation is unknown. In our patient, *SFTPB, SFTPC*, and *ABCA3* genes associated with congenital surfactant deficiency were analyzed by next-generation sequencing. Two variants were identified: likely pathogenic missense variant: c.604G>A (p.Gly202Arg) and splice region/intron variant c.4036-3C>G. The first variant has already been described in a patient with congenital surfactant deficiency in Iran [19]. The other one was not defined yet, but this variant can disturb RNA splicing resulting in the loss of exons or the inclusion of introns and an altered protein-coding sequence. Identification of homozygous or compound heterozygous *ABCA3* gene mutations, commonly predicts more challenging clinical profiles and poor outcomes compared to patients with a single *ABCA3* mutation and those with no defined genetic abnormalities [20], which is in agreement with the severe condition and lethal outcome of our patient.

The histopathological findings in the lungs of the patients with congenital surfactant deficiency consist of AEC2 hyperplasia, pulmonary alveolar proteinosis (PAP), inter-alveolar septa thickening, fibrosis, and inflammation. Frequently, prominent foamy macrophages are found in the airspaces, often embedded in proteinaceous material. Although the findings of routine examination may indicate one of the disorders, they cannot distinguish among different genetic causes since they are nonspecific for any of the Sp-B, Sp-C, and *ABCA3* mutations [21]. A molecular diagnosis is needed to determine the specific mutation affecting each case. Therefore, the histopathological examination was not performed in our patient’s case.

In the differential diagnosis of unclear neonatal respiratory failure, we need to take into consideration also other rare causes of respiratory distress. Among them are lethal lung developmental disorders, which result from abnormal or suppressed lung development during the fetal period. In this group, we distinguish alveolar capillary dysplasia with misalignment of the pulmonary veins (ACDMPV), acinar dysplasia (AcDys), congenital alveolar dysplasia (CAD), and pulmonary hypoplasia (PH) [10].

Another rare cause of neonatal respiratory failure might be congenital central hypoventilation syndrome, also known as Ondine’s Curse, caused by a mutation in the *PHOX2B* gene and characterized by severely impaired central autonomic control of breathing and dysfunction of the autonomous nervous system [10,22].

Apart from pulmonary causes, it is crucial to exclude any otolaryngeal congenital disabilities. In our patient, for some time, the tracheoesophageal fistula was suspected. Bronchoscopy suggested the presence of a large canal on the posterior tracheal wall. Additionally, the mucous membrane of the trachea was macerated, and many thrombi were found in the respiratory tract, which suggested destruction due to aspiration through the fistula. After administering total parenteral nutrition, we observed a recovery of the mucous membrane, which was first interpreted as a result of proper fistula management. Since the diagnosis was eventually ruled out, it seems that the injury of the mucous membrane was due to aggressive ventilation with very high ventilatory pressure, and it improved only when we managed to stabilize the patient and reduce the respiratory parameters. Throughout the diagnostic process in newborns with unknown causes of severe respiratory distress, it is crucial to be careful with the interpretation of endoscopic imagination. The worsening of the mucous membrane of the respiratory tract might result from various conditions, both anatomic or pathophysiologic.

It is also worth mentioning that we observed a lack of ventilation reserve in our patient, leading to fast heart arrest when extubation occurred.

As of today, there is no causal treatment for congenital surfactant deficiency. The starting point remains the management of respiratory distress by ventilation and oxygen therapy. Anti-inflammatory therapy with steroid course was described by Ciantelli M. et al. to be unbeneficial [23]. Our patient also received two courses of steroids, and they did not cause any positive effects. Nishida D. et al. reported an improvement after azithromycin administration, although the exact mechanism of this therapy is still unclear [24]. Some cases report improvement in patients with interstitial lung disease after administration of hydroxychloroquine [25,26]. We did not try any of these therapies on our patient. The only transient positive effect that we observed was after lung lavage with surfactant. It enabled us to reduce the ventilation parameters and finish HFO ventilation. Similar short-time results after whole lung lavages were also described in the literature [27]. On the contrary, Si et al. reported favorable clinical outcomes in 3 infants with neonatal respiratory distress and compound heterozygous variants in *ABCA3* treated with a 3-drug therapeutic regimen of monthly methylprednisolone pulses and daily azithromycin and hydroxychloroquine [28]. However, it is unknown whether the improvement was attributable to the medical regimen or the natural history of the lung disease. Further studies should focus on individualized, precision-based therapeutic regimens for patients with *ABCA3* deficiency.

For now, it seems that a lung transplant is the only definitive treatment for most infants with severe congenital surfactant deficiency [29]. It remains the only option for end-stage lung disease. Its 5-year survival was described to be approximately 50% [30]. Unfortunately, there is poor availability of lung transfer in such young patients, and the number of centers performing these surgeries is limited due to logistic problems and the complexity of these interventions. Many patients need to be gathered to gain enough technical expertise while finding donors requires a dedicated network and program. Moreover, surgical and medical issues, like managing immunosuppression and follow-up in infants, make lung and cardiac transplants challenging to accomplish [31]. Although rarely performed in Europe, lung transplants have been undertaken at St Louis Children’s Hospital (St. Louis, MO, USA). Between 1993 and 2015, 44 children were reported; 28 were transplanted at age <1 year, and 16 were older [32].

Recently, Forstner et al. used a human pulmonary epithelial cell line (A549) and machine-learning algorithms to develop a phenotypic assay to detect morphologic differences between stably transfected cells expressing wild-type *ABCA3* or missense variants. They next used this phenotypic assay to screen 1280 U.S. Food and Drug Administration (FDA)-approved small molecules and identified cyclosporine as a corrector of several *ABCA3* mutants that disrupt intracellular trafficking [33]. Hence, cyclosporin A may be selected for orphan drug evaluation in controlled repurposing trials in patients with *ABCA3* mutations [33,34].

Even though establishing the diagnosis does not alter the outcome of patients with congenital surfactant defects and frequently, the diagnosis is set after the patient’s death, it is essential to adequately counsel the parents and family members about the recurrence risk.

## 4. Conclusions

The patient’s clinical presentation is consistent with a case of respiratory failure due to surfactant deficiency. The histopathological findings in patients with surfactant deficiency are nonspecific for any SP-B, SP-C, and *ABCA*3 mutations, and a molecular diagnosis is needed. In the patient’s genetic analysis, two heterozygous *in trans* variants of the gene *ABCA3* were found. We conclude that they are responsible for this patient’s respiratory failure. Genetic counseling was suggested for the parents and relatives, informed of the recurrence risks.

Congenital surfactant defects are rare conditions. They must be considered among term newborns with respiratory failure of unclear etiology. Many unknown, unique possible variants in several genes encoding proteins are crucial in surfactant metabolism. Therefore, it is strongly recommended to use simultaneous, broad-spectrum approaches such as next-generation sequencing with panels targeted to surfactant-related genes or whole-exome sequencing [8,10].

## Figures and Tables

**Figure 1 diagnostics-12-01084-f001:**
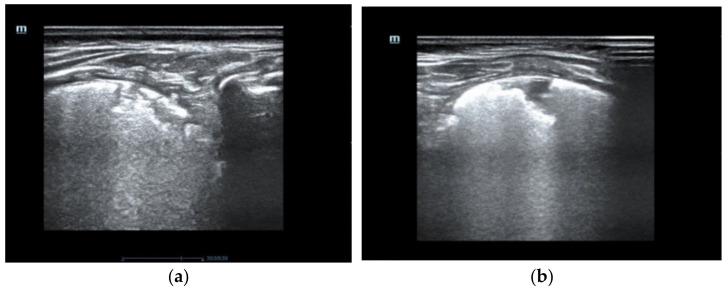
The pulmonary ultrasound shows compact B artifacts creating the image of white lungs with many subpleural consolidations. (**a**) posterior part of the right lung (**b**) the rear part of the left lung.

**Figure 2 diagnostics-12-01084-f002:**
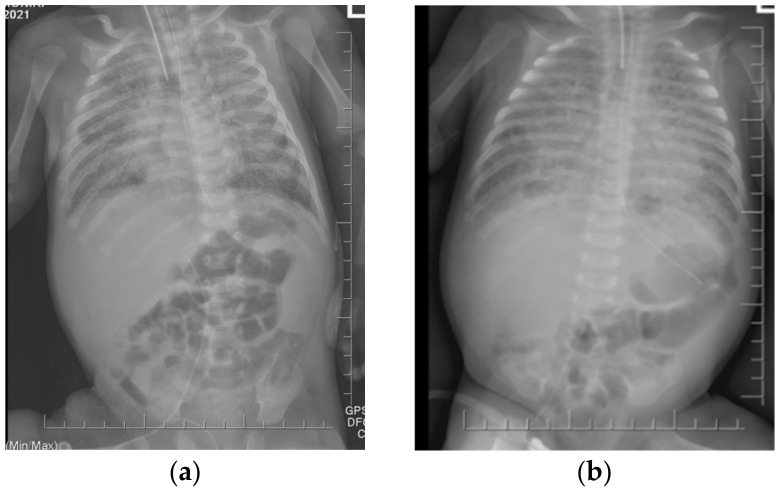
The chest X-rays revealed diffuse opacification of both lungs with air bronchograms. No improvement has been observed during one month. (**a**) 54th day of life (**b**) 85th day of life.

**Figure 3 diagnostics-12-01084-f003:**
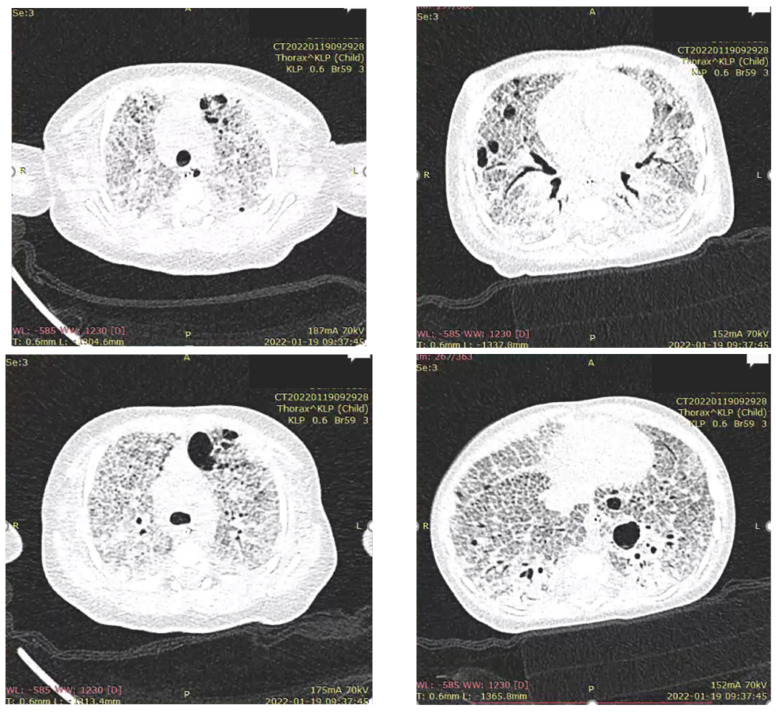
Computed tomography shows a characteristic pattern of interstitial lung disease with ground-glass opacities and pneumatocele.

**Figure 4 diagnostics-12-01084-f004:**
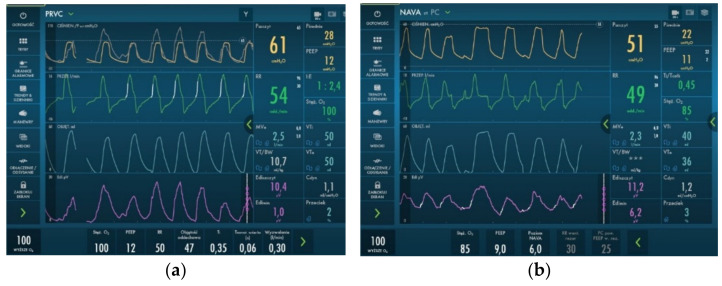
Invasive ventilation parameters used during hospitalization. We observed a reduction in PIP and MAP parameters after introducing NAVA ventilation. (**a**) PRVC ventilation; (**b**) NAVA ventilation.

## Data Availability

Not applicable.

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
