# Peer review of "Diagnostic Challenges in Neonatal Respiratory Distress—Congenital Surfactant Metabolism Dysfunction Caused by ABCA3 Mutation"

_diagnostics, 2022, doi:10.3390/diagnostics12051084_

Round 1
Reviewer 1 Report
I have read the manuscript titled "Diagnostic challenges in neonatal respiratory distress – congenital surfactant metabolism dysfunction caused by ABCA3 mutation" by Dr. Rogulska and coworkers.
It is a case report about a late preterm infant with a congenital surfactant deficiency due to ABCA3 gen deficiency, with an extensive and very nice description of the patient's admission during 99 days of life in their unit, the challenging diagnosis and treatments received. The case is very properly written, and can be of interest to many neonatologists or paediatric pneumologists.
I only suggest the following minor revisions before I recommend the publication of this report:
Commentary 1: Most of the references are not recent, as they were published before 2015. Although the references are scarce in this issue, there are more recent publications about ABCA3 deficiency in newborns, as well as some progress towards specific treatment that should be cited: Wambach et al. Am J Respir Cell Mol Biol 2022; or Wang et al. Transl Pediatr 2021. In the case of previous similar case reports, there are also some examples of more recent references: Si et al. J Pediatr 2021; Hu et al. Hum Mutat 2020; López Castillo et al. Arch Bronconeumol (Engl Ed) 2018.Commentary 2: The discussion is too long, the paragraphs that repeat the introduction about congenital deficiencies in surfactant should be removed.
Commentary 3: The images shown in figure 1 do not correspond to “completely atelectatic white lungs” (lines 102-103, pag 3), the legend that appears in figure 1 (lines 108-109) describes better the images included: “compact B artifacts creating the image of white lungs with many subpleural consolidations.”
Author Response
Thank you very much for reading our paper and for your comments. We are glad that you found it interesting and worth publishing and we introduced the changes according to your suggestions.
- We included the citations of papers that you mentioned and some important points that they provide. We updated the references.
185-188 - Qualitative functional characterization of ABCA3 missense variants suggests 2 pathogenic classes: disrupted intracellular trafficking (type I mutant) or impaired ATPase-mediated phospholipid transport into the lamellar bodies (type II mutant). Each disease-associated ABCA3 variant is associated with a diverse pulmonary phenotype [Hu et al].
208-212 - Identification of homozygous or compound heterozygous ABCA3 gene mutations, commonly predicts more challenging clinical profiles and poor outcomes compared to patients with a single ABCA3 mutation and those with no defined genetic abnormalities [Wang et al], which is in agreement with the severe condition and lethal outcome of our patient.
259-265 - On the contrary, Si et al. reported favorable clinical outcomes in 3 infants with neonatal respiratory distress and compound heterozygous variants in ABCA3 treated with a 3-drug therapeutic regimen of monthly methylprednisolone pulses and daily azithromycin and hydroxychloroquine [Si et al]. However, it is unknown whether the improvement was attributable to the medical regimen or the natural history of the lung disease. Further studies should focus on individualized, precision-based therapeutic regimens for patients with ABCA3 deficiency.
278-285 - Recently, Forstner et al. used a human pulmonary epithelial cell line (A549) and machine-learning algorithms to develop a phenotypic assay to detect morphologic differences between stably transfected cells expressing wild-type ABCA3 or missense variants. They next used this phenotypic assay to screen 1,280 U.S. Food and Drug Administration (FDA)-approved small molecules and identified cyclosporine as a corrector of several ABCA3 mutants that disrupt intracellular trafficking. Hence, cyclosporin A may be selected for orphan drug evaluation in controlled repurposing trials in patients with ABCA3 mutations [Forstner et al, Wambach et al].
- We removed from the Discussion the paragraphs that repeat the introduction about congenital deficiencies in surfactant. However, according to your suggestions, we added some information about treatment and diagnostics, so in fact, it did not shorten the discussion much.
- We changed the description of the image (lines 102-103). Now it is: …compact B-artifacts create the image of white lungs with many subpleural consolidations.
Reviewer 2 Report
The authors describe an interesting case of congenital surfactant metabolism dysfunction caused by ABCA3 mutation in a newborn with neonatal respiratory distress.
The article is easy to understand despite the topic and written with adequate references and comparisons to literature. Moreover, it focuses on a fascinating field of pediatric pulmonology: Inborn errors of pulmonary surfactant metabolism.
In my opinion, the article could be accepted for publication in this present form.
Author Response
Thank you very much for reading our paper and for your positive comments. We are glad that you found it interesting and ready for publication.